# Recent Advancements and Trends in Postharvest Application of Edible Coatings on Bananas: A Comprehensive Review

**DOI:** 10.3390/plants14040581

**Published:** 2025-02-14

**Authors:** Mawande H. Shinga, Yardjouma Silue, Olaniyi A. Fawole

**Affiliations:** 1Postharvest and Agroprocessing Research Centre, Department of Botany and Plant Biotechnology, University of Johannesburg, P.O. Box 524, Auckland Park, Johannesburg 2006, South Africa; shingamh31@gmail.com (M.H.S.); siluey@uj.ac.za (Y.S.); 2South African Research Chairs Initiative in Sustainable Preservation and Agroprocessing Research, Faculty of Science, University of Johannesburg, P.O. Box 524, Auckland Park, Johannesburg 2006, South Africa

**Keywords:** bibliometric analysis, edible coatings, banana, active ingredient, ripening, mode of action

## Abstract

Bananas (*Musa* spp.) are among the most widely consumed fruits globally, yet their high perishability and short shelf-life pose significant challenges to the postharvest industry. To address this, edible coatings have been extensively studied for their ability to preserve the physical, microbiological, and sensory qualities of bananas. Among various types of edible coatings, polysaccharide-based coatings, particularly chitosan, have emerged as the most effective. The dipping method is predominantly employed for their application, surpassing spraying and brushing techniques. This review integrates insights from bibliometric analysis using Scopus, revealing that research on edible coatings for bananas began in 2009, with 45 journals contributing to the field. Key trends, including publication growth, author contributions, and geographical focus, are explored through VOS-viewer analysis. Mechanistically, edible coatings enhance postharvest banana quality by limiting gaseous exchange, reducing water loss, and preventing lipid migration. Performance is further improved by incorporating active ingredients such as antioxidants, antimicrobials, and plasticizers. Despite their benefits over synthetic chemicals, the commercial adoption of edible coatings faces limitations, related to scalability and practicality. This review highlights these challenges while proposing future directions for advancing edible coating technologies for banana preservation.

## 1. Introduction

Bananas (*Musa* spp.), belonging to the family Musaceae, are among the most essential fruits in tropical regions and are widely consumed due to their affordability and high nutritional value. They serve as a significant source of carbohydrates, vitamins, fibers, and phenolic compounds [1,2,3]. In 2022, global banana production reached 135 million tons, with India leading production, followed by China and Brazil. Bananas also dominate the global fruit trade in terms of export value [4,5]. Due to their climacteric nature, bananas are harvested at the green stage and undergo three physiological stages, which include pre-climacteric (green stage), climacteric (ripening stage), and senescence (over-ripening and decay) [6,7,8]. During ripening, bananas undergo various biochemical changes, including conversion of starch to sugars, formation of aromatic compounds, enzymatic breakdown of the cell wall, and chlorophyll degradation, which reveals carotenoid pigments that give the fruit its yellow color [6,9,10]. While these changes enhance the sensory and nutritional qualities of bananas, they significantly reduce shelf life and marketability, posing a challenge to the postharvest industry.

The short shelf life of bananas is a global issue due to the complexity of the supply chain, which includes production, transportation, handling, storage, packaging, and distribution [1,11]. Postharvest losses range from 25% to 50%, primarily due to physiological changes, tissue softening, microbial spoilage, and improper handling during transportation and storage [12,13]. High storage temperatures and humidity further exacerbate losses, potentially resulting in up to 40% fruit spoilage.

To mitigate postharvest losses, various technologies have been explored, including controlled atmosphere storage, low-temperature storage, ethylene antagonists, and surface coatings [14,15,16,17]. However, these methods have limitations, such as high costs, chilling injuries, and uneven ripening. Among these, edible coatings have emerged as a promising alternative due to their eco-friendliness, cost-effectiveness, safety, and additional properties such as antioxidant and antimicrobial activity [18,19]. Edible coatings can be categorized into lipid-based, polysaccharide-based, and protein-based coatings. They may also be applied as composites or in a layer-by-layer approach [20,21,22,23,24,25,26]. Extensive research has demonstrated the effectiveness of edible coatings in preserving bananas by protecting them from mechanical and microbial damage, delaying deterioration, retaining volatiles, and improving their appearance [27,28,29,30,31,32]. Furthermore, edible coatings align with consumer preferences for safe, sustainable food products, offering a viable alternative to synthetic coatings.

Despite their widespread application, a significant gap persists in the literature, particularly in the integration of systematic reviews with bibliometric analysis. Existing reviews [1,33] provide valuable insights but are often limited by selective criteria, potentially overlooking emerging trends and interdisciplinary connections. Bibliometric analysis offers a data-driven approach to systematically map the conceptual evolution of edible coating technologies in fruit preservation [34,35,36]. This method reveals key research themes and tracks scientific developments through indicators such as influential authors, countries, journals, academic affiliations, and research collaborations [35,37]. Integrating bibliometric analysis with a systematic review provides researchers with a comprehensive understanding of the field, uncovering underlying patterns, research gaps, and future directions.

This review aims to address these gaps by combining systematic and bibliometric analyses to explore the advancements in edible coating technologies for bananas. It seeks to (1) examine the evolution of scientific production in this field, (2) identify key contributors and research trends, and (3) highlight future directions for enhancing edible coating technologies for banana preservation.

## 2. Materials and Methods

The methodology for this study was accurately designed to ensure a comprehensive systematic and bibliometric analysis. The process began with clearly defining the study’s scope and aim. The scope was broad to cover all relevant aspects of research on edible coatings used for banana fruit preservation. This included, but was not limited to, the materials used, their effectiveness in banana preservation, mode of action, sustainability impacts, and various application techniques. Based on this scope, selecting a suitable database for data collection was crucial. The Scopus database (https://www.scopus.com/, accessed on 3 June 2024) was selected due to its extensive and comprehensive coverage of scientific publications. Scopus is a widely recognized repository esteemed for its extensive coverage and reliable content [38]. Its wide-ranging collection of literature encompasses diverse fields, making it an ideal resource for investigating multidisciplinary topics such as edible coatings.

### 2.1. Data Collection

Data collection followed the Preferred Reporting Items for Systematic Reviews and Meta-Analyses (PRISMA) guidelines [39], a widely used framework that ensures transparency and reproducibility in systematic reviews. The PRISMA process involved four key stages: Identification, Screening, Eligibility, and Inclusion.

#### 2.1.1. Identification

Data collection began on 3 June 2024, using the Scopus database, recognized for its wide-ranging academic publication coverage. The search strategy utilised a set of keywords combined using Boolean operators (OR and AND), truncation symbols (*), brackets, and quotation marks (“…”) to ensure thorough and relevant results. The specific search was as follows: “edible coating* AND (banana OR *Musa* spp.). The obtained documents were screened using the settings “limited to articles” and “limited to English”. This approach maximizes the relevance and quality of the search results while maintaining a manageable and interpretable dataset.

#### 2.1.2. Inclusion and Exclusion Criteria

The search resulted in a diverse range of documents (115), including research articles (87), review articles (7), book chapters (1), and conference proceedings (16), written in different languages (3). Given that the goal of this study is to provide a clear picture of the latest advances and discoveries, evaluate scientific production, and analyze the research trends related to the topic, it is pivotal to focus on primary research data without any interpretation and synthesis. Therefore, the original articles that were written in English and reached the final stage of the publication process were considered for analysis. This step was crucial in ensuring the analysis was grounded in the most relevant literature. Since this step is important for accurate and comprehensive data, a manual screening was performed to exclude documents that did not specifically pertain to edible coatings on whole banana fruit preservation or did not comply with the criteria. In addition, the fresh-cut form was excluded because of its potential difference in physiological responses, coating requirements, and mechanisms compared to whole bananas. The documents deemed eligible based on our inclusion criteria were recorded for further analysis.

### 2.2. Data Preprocessing and Analytical Tools

The collected data underwent preprocessing to ensure consistency and accuracy. Microsoft Excel v. 2308 was used for data cleaning, which involved removing duplicate records, consolidating synonyms (e.g., “gum Arabic” vs. “Arabic gum”), and standardizing keyword variations (e.g., “edible coating” vs. “edible coatings”). Singular and plural inconsistencies were also addressed. For bibliometric analysis, R Studio (Bibliometric v. 4.1.3 package) was used to examine key contributors, research trends, and global statistics on edible coatings for bananas [40,41,42,43]. VOSviewer v. 1.6.19 was employed to visualize networks of keywords, co-authors, and research collaborations. This dual approach enabled a comprehensive mapping of the research landscape.

## 3. Results

### 3.1. The Research Performance Statistics on Edible Coatings for Banana Fruit

The application of edible coatings for banana preservation has demonstrated steady growth in scientific interest over the years. Figure 1 shows that between 2009 and 2024, 57 original research articles were published, authored by 217 contributors across 45 unique sources. The annual growth rate of publications in this domain is 13.85%, reflecting a positive but comparatively slower expansion than the broader field of edible coatings for fruit preservation, with an average growth rate of 16.94% [44]. This suggests that the application of edible coatings for bananas remains underexplored relative to other fruits, presenting opportunities for further investigation.

### 3.2. Publication and Citation Trends

The research trends on edible coatings (ECs) for banana preservation can be divided into two distinct phases: an initial decade of irregular growth (2009–2019) and a phase of rapid expansion (2019–2024), as shown in Figure 2. During the first decade, research activity was sporadic, with no publications recorded in 2012 and 2013. The annual output during this period ranged from one to five publications, reflecting limited growth likely due to a lack of widespread awareness about edible coatings, minimal technological advancements, and reduced focus on sustainability-driven postharvest solutions. In contrast, the second phase, starting in 2019, marked a significant surge in research activity, with publication output peaking at eight articles in both 2021 and 2022. Although the number of publications in 2024 is slightly below this peak, the trend suggests promising future growth in this area. This increase in publications after 2019 may be attributed to several factors, including growing consumer awareness of biodegradable and eco-friendly preservation methods, advancements in edible coating technologies (e.g., nanotechnology), and global initiatives aimed at reducing food waste and promoting sustainability in the food industry.

Citation trends closely mirror the growth in publications, indicating the increasing recognition of the importance of edible coatings in addressing postharvest losses for bananas. The heightened research activity in recent years underscores the relevance and impact of this field, particularly during the rapid expansion phase. Moreover, these trends in edible coating research for bananas align with the broader patterns observed in edible coatings for fruit preservation in general [44]. However, the delayed growth in banana-specific studies during the initial decade highlights a previously underexplored area compared to research on other fruits. These findings suggest that while progress has been made, there remains substantial potential for further innovation and exploration in the application of edible coatings for banana preservation.

### 3.3. Network Visualization of Academic Journals of Edible Coatings on Banana Fruit

Figure 3 presents a network analysis of academic journals focused on edible coatings for banana preservation. The visualization highlights the interconnectedness of scientific publications in this research domain, identifying key journals and their role in disseminating and integrating research findings. The analysis reveals that leading journals such as the *International Journal of Biological Macromolecules*, *LWT*, *Journal of Food Processing and Preservation*, and *Scientia Horticulturae* are pivotal in promoting collaboration and knowledge exchange. These journals frequently reference each other, emphasizing a strong overlap in research topics, including edible materials, biopolymers, postharvest management, and their applications in fruits. The frequent citation of articles among these journals underscores their collective contribution to advancing edible coatings as a sustainable solution for banana fruit preservation.

The network visualization provides additional insights into the flow of knowledge within this field. Each node in the network represents a journal, with the size of the node corresponding to the number of publications, highlighting the most productive sources of research. Larger nodes, such as those representing the *International Journal of Biological Macromolecules* and *LWT*, indicate their significant role in driving research activity.

Table 1 highlights the five most relevant journals contributing to research on edible coatings for postharvest banana preservation. Together, these journals published 14 articles, representing approximately 25% of the total 57 documents retrieved in this study. This concentration of publications underscores the central role these journals play in advancing the field. The high productivity of certain journals is often linked to their scope, which typically covers international topics of interest, encouraging researchers to contribute to these widely recognized platforms. Notably, the disparity in impact factors between the *International Journal of Biological Macromolecules* (8.2) and the *Journal of Food Processing and Preservation* (2.6) suggests that journal productivity is more closely tied to the journal’s scope than its impact factor.

Leading journals in this field serve as crucial platforms for disseminating original research and fostering theoretical advancements. For instance, the *International Journal of Biological Macromolecules*, one of the three most productive journals, focuses extensively on the chemical and biological properties of materials used in postharvest technologies, including edible coatings. This emphasis aligns with the research needs of edible coatings for bananas, where biopolymer composition and functionality play critical roles. Similarly, the *Journal of Food Processing and Preservation* centers on emerging technologies and advances in food preservation, addressing chemical, physical, quality, and engineering properties of food materials. *LWT*, another prominent journal, focuses on food processing, preservation, and distribution, making it a natural fit for research on edible coatings to improve the shelf life of climacteric fruits like bananas.

### 3.4. Geographic Distribution and Global Collaboration Network in Edible Coatings for Banana Fruit Preservation

The global research landscape on edible coatings for banana fruit preservation spans contributions from 27 countries, reflecting the field’s broad relevance, ongoing development, and significant global engagement (Figure 4A). Asia emerges as the leading region in terms of research output, with China and Malaysia at the forefront. This prominence can be attributed to their major roles in banana production and export, which necessitate innovative approaches to enhance fruit quality and shelf life. Africa is the second most significant region, driven by the essential role of bananas in ensuring food security and generating income through exports. These regional contributions underline the socioeconomic importance of bananas in these areas and the need for effective postharvest preservation strategies [45].

The collaboration map in Figure 4B further illustrates the global nature of research on edible coatings for banana preservation, revealing extensive international partnerships. Countries such as China, India, and the United States play pivotal roles in fostering global collaborations, reflecting the shared recognition of edible coatings as a sustainable solution for reducing postharvest losses. These collaborations align with the United Nations Sustainable Development Goals (SDG 17), which emphasize global partnerships to tackle pressing challenges, including food security and sustainability. The interconnected research efforts highlight the importance of edible coatings in addressing these critical issues and advancing toward global goals.

## 4. Edible Coatings: Materials-Based Coatings and Methods

Edible coatings are thin layers composed of either chemical or biological substances applied to the surface of food products [46,47]. Their primary function is to reduce gaseous exchange, thereby slowing down ripening, improving product quality, and extending shelf life (Figure 5). According to Mitelut et al. [47], edible coatings form a semi-permeable barrier that minimizes moisture loss and solute movement, effectively delaying spoilage and preserving freshness. Different types of edible coatings have been developed, including lipid-based, polysaccharide-based, protein-based, and their derivatives, each tailored for specific applications in food preservation [1,22]. Lipid-based coatings are widely used in meat products due to their hydrophobic properties and excellent moisture barrier capacity [20,21]. In contrast, polysaccharide- and protein-based coatings are predominantly applied to horticultural crops such as bananas, as they effectively regulate physiological responses and maintain fruit quality during storage.

Keyword analysis provides valuable insights into the research landscape of edible coatings, highlighting key themes and trends. Using multiple correspondence analysis (MCA) and keyword frequency analysis (Figure 6A), significant patterns in the field were identified. The most frequent keywords include *chitosan*, *shelf life*, *fruit ripening*, *gum Arabic*, *starch*, *carboxymethyl cellulose*, and *lemon extract*, demonstrating a strong emphasis on biopolymer-based coatings. This reflects the primary focus of research on developing materials that enhance banana shelf life while preserving fruit quality. Keywords such as *edible coatings* and *banana* were excluded from frequency analysis to focus on more specific topics within the study. Further analysis grouped these keywords into three clusters (Figure 6B), representing distinct research priorities in edible coatings. Cluster 1, represented in blue, focuses on keywords like *drop impact* and *maximum spread factor*, indicating research on droplet spreading behavior, particularly in the context of spray application methods. Cluster 2, shown in green, includes keywords such as *wettability* and *spreading coefficient*, emphasizing the exploration of interactions between liquids and surfaces, including coating viscosity and surface roughness, to optimize application efficiency. The largest cluster, represented in purple, highlights keywords related to materials-based coatings such as *chitosan*, *gum Arabic*, *starch*, and *carboxymethyl cellulose*, alongside physiological responses such as *respiration* and *ethylene*. This cluster also includes quality-related keywords like *shelf life*, *bioactive compounds*, and *lemon extract*, reflecting research on natural and sustainable materials designed to enhance banana shelf life by regulating physiological responses while preserving fruit quality. Overall, these findings emphasize the importance of continued research to address critical challenges in banana preservation, ensuring improved shelf life and reduced postharvest losses.

### 4.1. Polysaccharide-Based Coatings on Banana Fruit

Polysaccharide-based biopolymers have been extensively studied for their applications as edible coatings to preserve banana fruit, given their biodegradability and effectiveness in extending shelf life. Among these, chitosan stands out as the most researched material, followed by carboxymethyl cellulose (CMC) and gum Arabic (Table 2). Chitosan has gained recognition due to its film-forming ability, non-toxicity, biocompatibility, abundance, and capacity to act as a barrier to gaseous exchange [48,49,50,51]. Studies by Novianti and Dwivany [52] and Gol and Ramana Rao [53] demonstrated that chitosan-based coatings effectively delayed banana ripening by maintaining peel color and biochemical properties and minimizing weight loss at ambient temperatures. Additionally, chitosan was shown to suppress ethylene-related gene expression levels during early ripening stages [54]. However, some studies reported variable performance; for instance, Al-Qurashi et al. [55] found chitosan less effective in certain measured parameters compared to trans-resveratrol and glycine betaine, though it remained a strong candidate for enhancing fruit firmness and suppressing CO_2_ and ethylene production.

Starch, another common polysaccharide, is widely used in food preservation due to its filmogenic capacity, cost-effectiveness, and availability. Its linear component, amylose, is particularly suitable for making edible coatings [56,57]. Starch is a natural biopolymer present in various plants, including wheat, corn, rice, and potatoes [58]. Starch-based coatings are clear and suitable for horticultural products. Thakur et al. [27] demonstrated that a rice starch coating reinforced with sucrose esters improved the quality and extended the postharvest life of bananas stored at 20 ± 2 °C.

Similarly, carrageenan, a polysaccharide extracted from red algae, has been utilized for its selective permeability for gaseous exchange [59,60,61,62]. A study revealed that bananas coated with 1.5% carrageenan maintained better quality and extended shelf life when stored at 20 °C and 26 °C, with optimal results at lower temperatures [63].

CMC, derived from cellulose, offers properties such as gloss enhancement, moisture retention, and color preservation, making it ideal for banana preservation [18,64,65,66,67,68]. Ali et al. [9] reported that 1.5% CMC delayed decay incidence, slowed weight loss, and suppressed chlorophyll degradation by inhibiting key enzymes like Mg-dechelatase and peroxidase. These changes preserved higher chlorophyll levels, reduced carotenoid accumulation, and suppressed ethylene biosynthesis, contributing to extended shelf life under ambient conditions. Similar findings by Kim et al. [69] confirmed CMC’s efficacy, making it a valuable alternative for natural coatings.

Aloe vera, a tropical and subtropical plant, is widely recognized for its exceptional medicinal properties [70]. *Aloe vera* gel edible coating has recently been developed and can potentially extend the shelf-life of fresh fruits [71]. It offers a safe and environmentally friendly alternative to synthetic preservatives. *Aloe vera* gel (AVG)-based edible coatings have been demonstrated to prevent moisture loss and maintain firmness, control respiratory rate and maturation, delay oxidative browning, and reduce microbial proliferation in fruits, including bananas [72,73,74]. A study by Jodhani and Nataraj [75] evaluated the possible effect of *Aloe vera* gel as an edible coating on postharvest quality and shelf-life of banana during storage at 23 ± 1 °C. This study found that treated banana fruit displayed a reduction in weight loss and the decay rate and slowed down the organic acid breakdown and TSS accumulation. AVG edible coating extended the banana shelf life up to 9 d with no sign of disease incident. Recent research has shown that AVG edible coatings reduced weight loss, delayed TSS and TA changes, minimized decay, and inhibited peel color changes in two banana cultivars stored at 10 °C or 25 ± 2 °C [76]. The AVG coating did not significantly affect the taste or overall acceptability, although panelists preferred coated bananas. The combined effects of AVG coating and storage at 10 °C was the most effective treatment in maintaining banana quality. The limitation of this study is that it could not provide measurements on ethylene and CO_2_ production since banana is a climacteric fruit. Overall, AVG-coated bananas had an extended shelf life of 4 d more compared to uncoated.

Mucilage, a slimy, soluble fiber classified as a hydrocolloid, can be extracted from various natural sources, including cactus pear cladodes and fruit, linseed, and okra [77,78,79]. As a polysaccharide, mucilage has attracted considerable attention in the cosmetic, pharmaceutical, and food industries due to its functional and beneficial properties [80]. *Opuntia ficus indica* mucilage (OFIM), derived from cactus pear cladodes, has demonstrated significant potential as an edible coating for bananas. Studies have shown that OFIM effectively delays both cell wall and chlorophyll degradation, suppressing carotenoid accumulation and consequently slowing peel color changes compared to uncoated bananas. Furthermore, OFIM-treated bananas exhibited reduced firmness loss, minimized weight loss, and lower total soluble solids (TSS), carbon dioxide (CO_2_), and ethylene production [10]. These findings suggest that OFIM could serve as a sustainable and natural alternative to chemical fungicides for banana preservation, offering a promising solution for extending shelf life while maintaining fruit quality.

### 4.2. Protein-Based Edible Coatings on Banana Fruit

Protein-based edible coatings, composed of natural heteropolymers and made up of long chains of amino acids, have gained recognition for their excellent oxygen barrier properties [81]. When dissolved in water, protein molecules typically engage in ionic interactions or hydrogen bonding, contributing to their structural integrity and functionality as a coating material. Commonly used proteins in edible coatings include soy, gluten, whey, and zein, which are widely applied due to their mechanical strength and ability to provide a protective barrier [18]. Despite this limitation, these coatings offer a diverse range of physical and mechanical properties that can be tailored to meet the preservation needs of different food products [20,82]. Protein-based coatings are particularly valued for their low moisture barrier, good mechanical strength, and high gas permeability, enabling them to effectively regulate the internal atmosphere of packaged produce [22,83]. Notably, gelatin-based coatings have been used to preserve bananas stored at 25 °C. Gelatin-coated bananas exhibited delayed ripening, as evidenced by a slight reduction in weight loss, softening, acidity, and sugar accumulation during storage [3]. These findings highlight the potential of protein-based edible coatings as a viable solution for extending the shelf life of bananas while maintaining their quality attributes.

### 4.3. Lipid-Based Edible Coatings on Banana Fruit

Lipid-based edible coatings, composed of small hydrophobic molecules, are widely used in food preservation to prevent moisture loss and reduce water vapor transmission [84]. These coatings provide an excellent moisture barrier, which helps to mitigate physiological deterioration in food products [22]. Additionally, lipid-based coatings enhance the visual appeal of fruits by imparting a glossy, shiny finish, making them particularly attractive for commercial applications [85]. Common materials used in lipid-based coatings include natural waxes, petroleum-derived waxes, mineral oils, fatty acids, and resins [86]. Among these, waxes and shellac are the most commonly applied lipids, primarily due to their effectiveness in improving the glossiness and overall appearance of fruits [86]. While lipids offer exceptional moisture barrier properties, they generally exhibit poor mechanical strength. They are often combined with proteins and polysaccharides in composite edible coatings to address this limitation, resulting in improved functionality and durability [87]. For instance, Mladenoska [88] demonstrated that beeswax effectively prevented the deterioration of banana fruit, maintained higher ascorbic acid levels, and enhanced fruit appearance compared to uncoated bananas. These findings highlight the potential of lipid-based coatings, particularly when combined with other materials, as a practical solution for extending banana shelf life and improving postharvest quality.

### 4.4. Composite-Based Edible Coatings on Banana Fruit

Composite-based edible coatings or layer-by-layer coatings are regarded as an innovative solution to overcome the limitations of individual coating materials while harnessing their combined benefits [83,89,90]. These coatings integrate different components, such as lipids, proteins, and polysaccharides, to create a multifunctional barrier tailored for specific food preservation needs. For example, lipids, which have poor mechanical properties, can be enhanced by incorporating water-soluble proteins or polysaccharides, while the lipids, in turn, reduce the high moisture permeability of hydrocolloids [81]. In the case of bananas, composite coatings have shown promising results in maintaining postharvest quality. Studies on chitosan and chitosan gallate composite coatings have reported improvements in several physicochemical attributes, including reduced weight loss, better peel color retention, enhanced membrane stability index, firmer pulp, and balanced total soluble solids (TSS) and titratable acidity (TA) [91]. Additionally, these coatings positively influenced antioxidant properties such as total phenolic content and free radical scavenging capacity (FRSC) as measured by DPPH IC50 values. However, their effect on ascorbic acid and total flavonoid concentrations was negligible. These findings indicate that composite edible coatings offer a practical approach to optimizing banana preservation by combining the strengths of multiple materials, ensuring extended shelf life and improved fruit quality.

## 5. Functionalization of Edible Coatings

Edible coatings have proven themselves to be reliable in preserving food products. However, the performance and functionality of edible coatings can be influenced by various factors such as the composition of the coating, coating thickness, application method, environmental conditions, food surface interactions, mechanical properties, permeability, and additive or active ingredients [84]. This section will focus on additives such as antioxidants, antimicrobials, and plasticizers, which can play a crucial role in extending the shelf life and safety of food products.

### 5.1. Antioxidants

Oxidation plays a crucial role in food spoilage, leading to significant food waste yearly [92]. Incorporating plant-based substances, extracts, or oils (vegetative and essential oils) into edible coatings enhances their effectiveness and makes them more environmentally friendly [93]. These plant extracts can be obtained from various parts of the plants, including roots, leaves, fruit peel, and seeds. These plant parts are rich in antioxidant compounds that boost the efficiency of the coatings. Hence, Jodhani and Nataraj [75] have reported that Aloe vera gel (50%) reinforced with lemon peel extract (15%) significantly extended the shelf-life, reducing the quality losses in banana fruit during storage under 23 ± 1 °C. Lemon peel extract (LPE) is known for its antioxidant activity [94]. Chitosan infused with gallic acid slowed down TSS increase, weight loss, and color change and preserved titratable acidity total phenolic content in banana fruit [91].

### 5.2. Antimicrobials

The antimicrobial compounds in edible coatings prevent the fruit from decaying by inhibiting microbial growth. Plant extracts have been reported to possess antimicrobial [95,96] and anti-inflammatory properties [95]. Previous studies have demonstrated that active ingredients containing antimicrobial activities are effective against bananas. For instance, silver nanoparticles (AgNPs) were incorporated in agar to suppress postharvest rot diseases and enhance fruit storage life during ripening [97]. AgNPs have been reported to contain antibacterial activity [98]. A study by Odetayo et al. [99] revealed that chitosan nanoparticles were more effective against decay incidence. Chitosan nanoparticles have been demonstrated to retard decay in banana until 11 days of storage (25 ± 1 °C) [100]. Lemon peel extract (LPE) has been incorporated into Aloe vera gel for effectiveness against fungi, particularly *Colletotrichum musae*, which causes one of the main diseases of banana fruit (*anthracnose*). Furthermore, essential oils extracted from plants also exhibit antimicrobial properties [101], preventing microbial growth in food products [102]. However, to our knowledge, no work has been reported on essential oils incorporated with edible coatings on banana fruit.

### 5.3. Plasticizers

A plasticizer is a substance that is added to edible coatings to increase their flexibility, workability, and plasticity [103]. Plasticizers disperse throughout the biopolymer network through hydrogen bonding to disrupt intermolecular interactions between chains and replace polymer–polymer interactions with polymer–plasticizer interactions [103]. There are different types of plasticizers, including glycerol, sorbitol, and polyethylene glycol 400 (PEG-400), and their effectiveness depends on the composition and size [104,105,106]. Among these plasticizers, glycerol is the most commonly used plasticizer due to its low molecular weight, water solubility, and hydrophilic nature [107]. Therefore, to ensure complete hydration and solubility of the coating, various ranges of glycerol concentrations are added, although this may not be effective [2,9,27,61].

### 5.4. Optimization of Edible Coatings for Banana Preservation

Optimizing edible coatings is a critical step to enhance their properties and effectiveness, particularly because the performance of substances such as antioxidants, antimicrobials, and plasticizers is dose-dependent. Achieving the optimal concentration of these components is essential but often challenging. Several studies have focused on optimizing edible coatings to meet specific objectives for postharvest fresh produce preservation. Key factors to consider during optimization include the selection of suitable edible coating materials, formulation development, functional properties, application techniques, cost-effectiveness, and consumer acceptance. These factors ensure that the edible coatings not only preserve the quality of fresh produce but are also practical and appealing for commercial use. Optimization is commonly conducted using multivariate statistical techniques such as response surface methodology (RSM) [108], partial least squares regression (PLSR) [109], canonical correlation analysis (CCA) [110], multivariate analysis of variance (MANOVA) [111], and artificial neural networks (ANNs) [112]. Among these methods, RSM is the most widely employed. It involves fitting a polynomial equation to experimental data to identify optimal conditions [113]. In the case of banana preservation, RSM has been effectively used to determine the optimal concentration of coating components. For example, Malmiri et al. [114] successfully applied RSM to optimize the concentration of chitosan and glycerol for banana fruit preservation. The study found that the optimized concentrations improved the quality of bananas stored under conditions of 26 ± 2 °C and 40–50% relative humidity. These findings underscore the importance of optimization techniques in tailoring edible coatings to achieve maximum efficacy in postharvest quality maintenance and shelf-life extension.

### 5.5. Edible Coating Application Techniques

The most commonly used methods for applying fruit coatings are dipping, spraying, and brushing [115,116,117,118]. These application techniques are depicted in Figure 5. Although other techniques, such as fluidized bed and foaming, are available, they are less frequently used in both commercial and laboratory settings [116,119]. Since these techniques can have varying effects on the quality, safety, and shelf-life of fruits, this section compares their efficiency and suitability for different types of produce or experimental setups. This also lays out the types of application techniques that could help researchers identify gaps or trends in the literature.

The choice of deposition method depends on the type of food being coated, its surface characteristics, and the primary purpose of coating [116]. Applying coating solutions involves adhesion, including diffusion between the coating solution and the food’s surface [120]. The wetting stage is crucial when the coating solution has a high affinity for the product surface application time is reduced, allowing for spontaneous coating [116]. Additionally, it is important to consider the ripening patterns of the product before applying it, whether it is climacteric or non-climacteric [121]. Studies have demonstrated various coating deposition methods on food products to improve their quality, enhance their safety, and extend shelf-life [122,123,124]. The dipping method is the most used method for coating food products, involving immersing a food sample in a coating-forming dispersion [120]. This approach ensures that the substrate is immersed in the coating solution with a sufficient quantity of solution for wetting the substrate and allowing complete interaction between the substrate and the coating matrix to form a thin layer [125,126,127]. Additionally, this method is easy to adopt, and it saves time and labor [115]. Kellerman et al. [128] reported that the dipping method produces a more homogeneous coating, resulting in a better preservation effect. The dipping method has been reported effective in banana preservation during postharvest life [100,129,130,131,132].

The spraying method is also one of the most common coating application methods on food products [133]. The spraying method uses less coating than the dipping method and prevents cross-contamination by microorganisms [134]. The spraying method increases the liquid surface by forming droplets and dispersing them across the food surface with a set of nozzles [116]. They are divided into three types, namely air-assisted airless atomization, pressure atomization, and air-spray atomization. Air-assisted airless atomization effectively addresses issues with high viscosity and high solids coatings, as well as issues related to heating and using higher fluid pressures for atomizing viscous materials [135]. Pressure atomization uses pressure without air. Small-size nozzles are utilized to channel high-pressure energy, ensuring the coating’s surface tension and high viscosity are suitable for application on food products [136]. Air-spray atomization is characterized by the friction between the fluid and air, which accelerates and disrupts the fluid flow, causing the atomization. It is also the most cost-effective method among the spraying techniques [126]. The spraying method application has not been extensively explored for banana fruit; however, it has been reported to be effective in maintaining quality and prolonging shelf-life [27,137,138,139].

The brushing method is versatile and cost-effective, and it can be applied to food products of any shape and size [115]. This method is not extensively used since it is labor-intensive and can result in uneven brushing coating application.

## 6. Composite Edible Coatings for Bananas and Their Effect on Quality Attributes

Table 3 presents the application of various edible coatings in postharvest banana preservation, highlighting the efficacy of different coating types, including polysaccharide-based, protein-based, lipid-based, and composite coatings. The data revealed the most commonly used materials for banana preservation and demonstrated the comparable efficacy of the coatings in maintaining fruit quality. Additionally, the findings underscore the versatility of edible coatings, particularly their ability to incorporate various active ingredients, enhancing their functionality in extending shelf-life and preserving key attributes of bananas.

## 7. Effect of Edible Coatings on Postharvest Diseases, Disorders, and Decay in Banana Fruit

Bananas are highly susceptible to various postharvest diseases and decay, which contribute significantly to postharvest losses and limit their shelf life during storage [148,149]. Traditionally, synthetic fungicides such as benomyl and thiabendazole have been used to manage postharvest diseases [150]. However, growing health concerns regarding chemical residues in food and the environment have driven demand for safer, non-chemical alternatives. Edible coatings have emerged as a promising solution for reducing the incidence of postharvest diseases and decay in bananas, offering a sustainable and consumer-friendly alternative [75,139,148]. Research highlights the efficacy of edible coatings in suppressing the development of postharvest diseases caused by microorganisms. For instance, as summarized in Table 4, agar–agar coatings were shown to significantly reduce disease incidence and severity caused by *Colletotrichum musae* and *Fusarium moniliforme*, two major postharvest pathogens in bananas [139]. Similarly, gum Arabic combined with chitosan was effective in minimizing disease incidence caused by *Colletotrichum musae* [148]. *Aloe vera* gel combined with lemon peel extract also demonstrated strong antimicrobial activity, suppressing disease incidence caused by *Colletotrichum musae* [75]. These findings highlight the ability of edible coatings to serve as both physical barriers and carriers of bioactive compounds with antimicrobial properties. In addition to disease suppression, edible coatings have also proven effective in reducing decay during storage. Coatings such as carboxymethyl cellulose and *Opuntia ficus*-*indica* mucilage (OFIM) significantly reduced decay, maintaining the visual and structural integrity of bananas during postharvest storage [9,10]. *Aloe vera* combined with chitosan also reduced decay, further supporting its potential as a dual-purpose coating for disease suppression and decay prevention [99]. These results emphasize the role of edible coatings in managing both microbial threats and physiological decay, ensuring extended shelf life and improved fruit quality during storage.

## 8. Mechanism of Action of Edible Coatings on Bananas

Banana fruit ripening is a coordinated process that involves processes that lead to various molecular, biochemical, and physiological changes [151]. Due to its climacteric nature, banana undergoes rapid ripening. The understanding of the fruit ripening process itself and how edible coatings modulate it to extend the shelf life are interesting to numerous authors. Hence, various coatings have been studied to investigate their modes of action in modulating gene expression, enhancing the antioxidant activity of reactive oxygen species, and preserving the cell wall integrity.

### 8.1. Modulating Gene Expression

Recent studies have highlighted the role of edible coatings, particularly chitosan-based coatings, in modulating gene expression to delay the ripening process in bananas. Transcriptomic analysis conducted by Dwivany et al. [152] provided crucial insights into the mechanisms behind the ripening delay induced by chitosan coatings. RNA sequencing of banana pulp revealed that over 80% of the reads were mapped to genome references, identifying candidate genes involved in fruit ripening in response to the coating. This study significantly contributes to the development of improved postharvest management strategies for bananas. Further supporting evidence comes from Yamamoto et al. [153], who investigated the 1-aminocyclopropan-1-carboxylic acid oxidase (ACO) genes, which encode an enzyme responsible for converting 1-aminocyclopropan-1-carboxylic acid to ethylene in bananas. The study confirmed that chitosan coatings delayed the ripening process by modulating ethylene biosynthesis pathways. Similarly, research by Novianti and Dwivany [52] demonstrated that chitosan-coated bananas exhibited altered expression patterns of key ripening-related genes such as ACS1 and ACO1, which significantly contributed to the prolongation of the ripening process.

A study investigating the effect of different concentrations of chitosan and chitosan nanoparticles on the expression of MA-ACS1 and MA-ACO genes further revealed that treated bananas showed reduced gene expression compared to untreated samples [100]. This reduction was attributed to the ability of chitosan nanoparticles to form an effective barrier, thereby decreasing ethylene production. This mechanism was corroborated by Esyanti et al. [154], who noted that the suppression of ethylene production due to reduced MA-ACS1 and MA-ACO expression levels plays an important role in regulating ripening-related processes. These findings provide compelling evidence for the ability of chitosan-based coatings to modulate gene expression, thereby offering an effective strategy for extending the shelf life and maintaining the quality of bananas during storage.

### 8.2. Regulation of Reactive Oxygen Species (ROS)

Horticultural crops, including bananas, naturally produce reactive oxygen species (ROS) such as hydrogen peroxide (H_2_O_2_), singlet oxygen (O_2_), hydroxyl radicals (OH^−^), and superoxide anion (O_2_^−^) [155]. However, during ripening, the overproduction of ROS can lead to oxidative stress, causing damage to cell membranes through lipid peroxidation [156]. Research has demonstrated that edible coatings can play a vital role in modulating ROS levels and enhancing the antioxidant defense mechanisms in bananas, thus delaying ripening and extending shelf life. A study by Awad et al. [91] showed that chitosan (CH) coatings incorporated with gallic acid (GA) or chitosan gallate (CG) effectively maintained the activities of critical enzymes such as polygalacturonase (PG), peroxidase (POD), and polyphenol oxidase (PPO) in bananas. These coatings delayed ripening by reducing POD activity during early storage, although POD levels eventually increased with prolonged storage. Similarly, GA and salicylic acid treatments were found to significantly enhance antioxidant enzyme activity, particularly POD, after nine days of storage [143]. These treatments also stimulated the biosynthesis of additional key antioxidant enzymes, including PPO, phenylalanine ammonia-lyase (PAL), and β-1,3-glucanase, as well as superoxide dismutase (SOD), catalase (CAT), and ascorbate peroxidase (APX) [143,157]. The defensive roles of PPO and POD against pathogen attacks further underscore their importance in maintaining postharvest fruit quality [143,158]. The transition from fruit maturation to senescence is marked by a gradual oxidative shift, characterized by increased ROS production and the upregulation of antioxidant enzyme activities and their associated genes [159,160,161]. By applying edible coatings, the antioxidant defense system in bananas can be enhanced, as these coatings stimulate the biosynthesis of key enzymes, reduce oxidative stress, and mitigate the deleterious effects of ROS. This reinforces the role of edible coatings as an effective strategy to extend the postharvest life of bananas by modulating ROS levels and maintaining fruit quality.

### 8.3. Regulation of Cell Wall Enzyme Activities

The degradation and disassembly of cell wall polysaccharides largely cause rapid ripening accompanied by fruit softening [9]. This presents a significant challenge in extending the storage life of the fruit [162,163,164,165]. The mechanism of ripening, which was correlated with biochemical processes, is presented in Figure 7. The diagram displays that the continuous production of ethylene causes a rapid change in color, aroma production, and softening occurrence, though other signal molecules are also involved in this process. A study by Ali et al. [9] demonstrated the efficacy of CMC in delaying banana softening during storage. The findings revealed that CMC significantly slowed down the activities of chlorophyllase, pheophytinase, Mg-dechelatase, and chlorophyll (Chl)-degrading peroxidase (Chl-POD), which consequently delayed color change during ripening. The delayed ripening results from chlorophyll degradation and catabolism through enzymatic processes, in which the chlorophyllase enzyme removes phytol from chlorophyllide [166]. Then, Mg-dechelatase further removes Mg from the nucleus of chlorophyllide; this conversion causes the breakdown of the chlorophyll ring structure, resulting in the degradation of the green color [10,167]. Additionally, Chl-POD directly breaks down the Chl-a pigment, and pheophytinase dephytylates pheophytin-a, eventually resulting in chlorophyll degradation [168,169,170]. The study also reported that CMC coatings suppressed fruit softening, as evidenced by higher retention of structural components such as cellulose, sodium carbonate-soluble pectin, chelate-soluble pectin, hemicellulose, and protopectin. These coatings simultaneously reduced water-soluble pectin levels, which are associated with fruit firmness loss during ripening [9].

The depolymerization of cell wall polysaccharides loosens the cell wall [171]. The involvement of key enzymes such as pectin methylesterase, polygalacturonase, β-glucosidase, and cellulase activities also influences the depolymerization process [9,172]. Shinga and Fawole [10] investigated the role of protopectin in maintaining fruit firmness and its degradation during ripening. Their study revealed that protopectin, a key component of the cell wall, breaks down into water-soluble pectin as the fruit ripens, a process accelerated by the action of the enzyme pectinesterase. The application of OFIM edible coating, as explored in the study, was highly effective in preserving fruit firmness. This was attributed to the coating’s high calcium content, which interacts with pectic acid in the cell wall to form calcium pectate, a compound that strengthens cell wall structure. This mechanism was further supported by Jobert et al. [173], who demonstrated the critical role of calcium in stabilizing the cell wall structure. The findings suggest that edible coatings like OFIM delay cell wall degradation and regulate the activity of softening-related enzymes in bananas during postharvest storage. By minimizing the breakdown of protopectin and forming stronger cell wall structures, these coatings effectively suppress fruit softening and help maintain firmness, thereby extending the shelf life of bananas and improving their overall quality during storage.

## 9. Limitations and Future Research

Edible coatings represent a promising technology for enhancing banana fruit quality and extending shelf life [22,67]. Numerous studies have explored their effectiveness, focusing largely on materials-based coatings and the incorporation of functional substances such as antioxidants and antimicrobials. Although these substances enhance the properties of the edible coatings, optimizing their incorporation strategies still requires attention.

In addition, several critical areas remain underexplored, such as the precise mechanisms by which edible coatings delay ripening, suppress microbial activity, and regulate physiological processes. Additionally, laboratory-scale studies have demonstrated their effectiveness, but there is a significant gap in evaluating their performance and scalability at pilot and commercial levels. This limitation hinders the transition of edible coating technologies from research to industry-wide application. A key factor determining industry acceptance of the new technology is economic feasibility, especially in low-income and resource-limited regions. While edible coatings can lower banana postharvest losses, there is still a lack of information on the cost of raw materials, processes (formulation and application methods), and return on investment, highlighting the lack of studies integrating techno-economic assessments.

Another notable gap is the lack of comprehensive studies on consumer acceptance, particularly regarding sensory attributes and the impact of coatings on taste, aroma, and texture. Edible coatings are promoted as safe and sustainable alternatives, but there is insufficient research on their biodegradability and environmental impacts, such as carbon footprint, waste management, and life cycle analysis. Addressing these factors is critical to validate their sustainability claims.

Despite their recognition as GRAS (Generally Recognized As Safe) by the US FDA (Food and Drug Administration and EFSA (European Food Security Authority), one of the most important constraining factors for the commercial implementation of edible coatings in banana industries is the lack of global regulatory standards and safety guidelines, as achieving compliance with food safety protocols can be complex and time-consuming, while the banana market is highly predominated by export (cross-border commercialization). This is even more constraining when the coating integrates active compounds such as antimicrobials, antioxidants, and nanomaterials. Indeed, the toxicity and potential long-term health impacts of certain functionalizing substances warrant thorough investigation to ensure consumer safety and regulatory approval.

To overcome these limitations, future research should prioritize the development of green, readily available, and cost-effective coating materials. Innovations such as controlled-release systems, optimized formulations, and novel application techniques should be explored to enhance functionality and adaptability. Mechanistic studies on how edible coatings suppress rapid ripening, regulate gas exchange, and delay enzymatic activities are crucial to refine their design and application further. Moreover, sensory evaluations, particularly about volatile compounds, should be conducted to ensure consumer acceptance and satisfaction. Research should also focus on improving the technological readiness of edible coatings for commercial-scale applications, including the development of efficient and scalable coating equipment. Lastly, a science-based approach to life cycle analysis, biodegradability assessment, and cost–benefit studies will facilitate the global adoption of edible coatings in the banana fruit industry. These steps will ensure that edible coating remains a viable solution for postharvest management and align with sustainability and consumer preferences.

## 10. Conclusions

Edible coatings have a significant potential to extend the shelf-life of banana fruit by slowing down the ripening process and minimizing quality degradation. This study analyzed a total of 57 documents published from 2009 to 2024 in Scopus. This study presents the current state of trends and identifies future research tendencies. The review findings are relevant to researchers actively applying edible coatings on banana fruit. Furthermore, polysaccharide-based edible coatings are extensively documented in the literature and show great promise when applied to banana fruit. Applying composite coatings has been demonstrated to enhance quality attributes due to improved barrier or mechanical properties, reduce transpiration rates, and improve postharvest storability. Additionally, incorporating active ingredients, including antioxidant agents, antimicrobial agents, and plasticizers, can further enhance the functionality of the coatings. The literature has indicated that banana fruit responds positively to edible coatings, suggesting that this technique approach offers a sustainable, cost-effective, and natural method for the postharvest management of banana fruit. Research has been conducted to optimize the formulation of edible coatings for banana fruit. However, no study has addressed the commercial feasibility of applying edible coatings on bananas or examined their real-life, commercial-scale trials. This study provides a comprehensive overview of the scientific literature on this research area, making it a valuable resource for researchers in this field.

## Figures and Tables

**Figure 1 plants-14-00581-f001:**
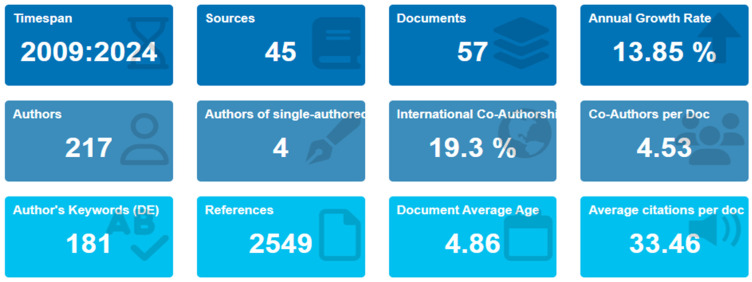
Global performance statistics of research on edible coatings for banana fruit preservation.

**Figure 2 plants-14-00581-f002:**
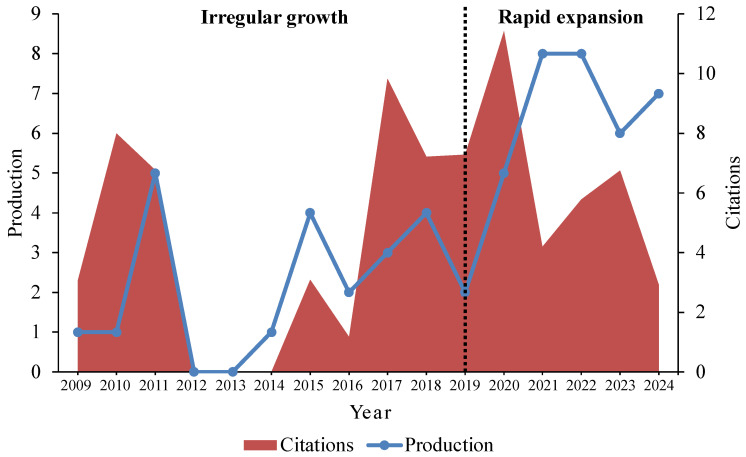
Annual publication and citation trends of edible coatings for banana fruit preservation based on the Scopus database from 2009 to 2024.

**Figure 3 plants-14-00581-f003:**
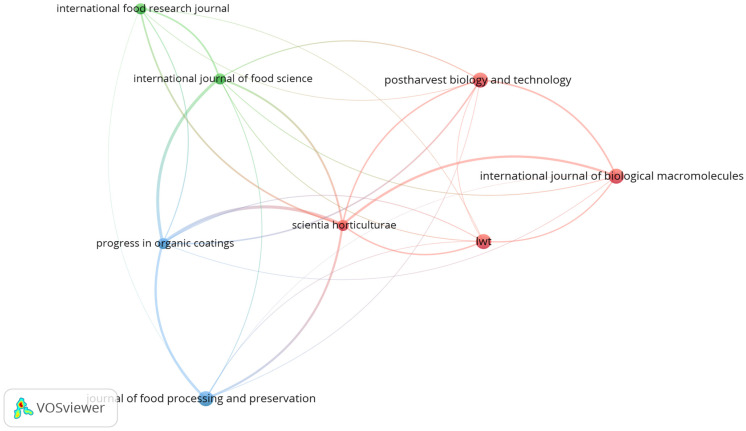
Map network visualization of research journals in edible coatings for banana fruit preservation (2009–2024), based on the data from the Scopus database. Each node in a network represents a journal, and the size of the node corresponds to the number of publications in that journal. The larger the node, the higher the publication count. The links between journals represent the bibliographic connection. The color represents clusters or groups of related journals. The connections between nodes represent bibliographic linkages, illustrating how research findings from one journal influence and contribute to work published in others. The clusters indicated by distinct colors represent groups of closely related journals, showcasing their thematic alignment and collaborative potential.

**Figure 4 plants-14-00581-f004:**
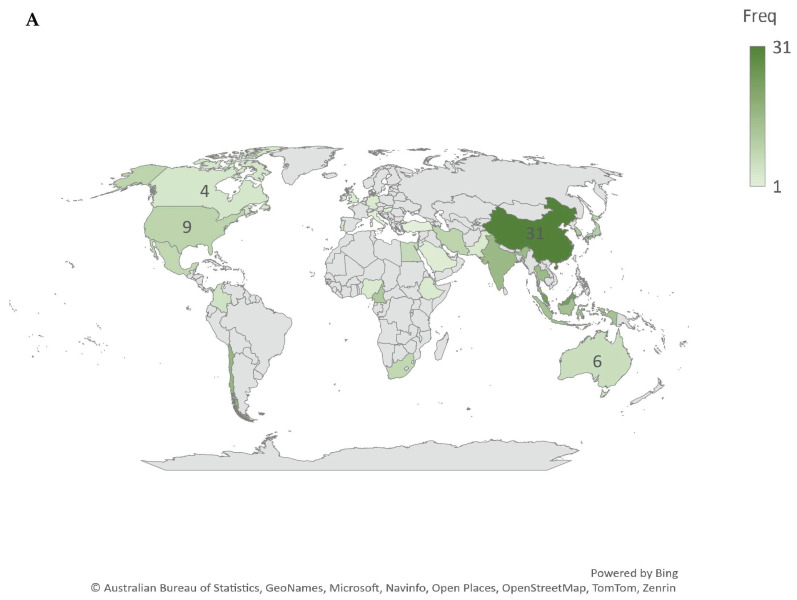
Map illustrating scientific production in the field of edible coatings for banana fruit preservation (**A**) and the global collaboration network of countries through bibliographic coupling of retrieved documents from the Scopus database (2009–2024) (**B**). In (**A**), each country is shaded according to the frequency of publications, with darker shades representing higher research output. In (**B**), each country is depicted as a node, with the size of the node corresponding to the number of publications from that country. The thickness of the lines connecting nodes represents the strength of collaborative links, while the proximity of nodes indicates the frequency of co-occurrence in publications. Distinct colors in (**B**) represent clusters of countries with robust collaborative research networks, highlighting regional and international partnerships in the field.

**Figure 5 plants-14-00581-f005:**
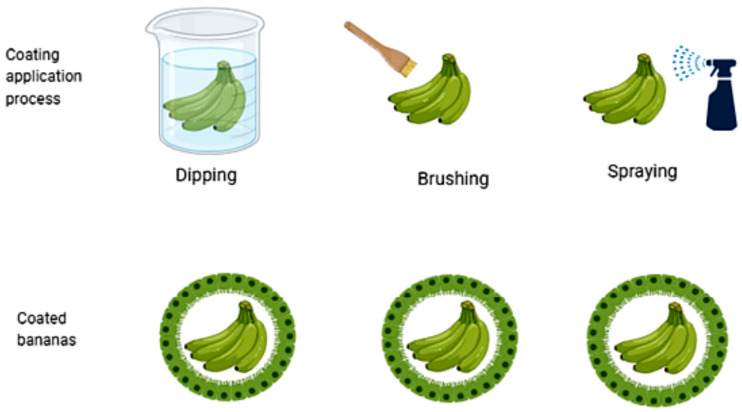
Application methods for edible coatings on banana fruit.

**Figure 6 plants-14-00581-f006:**
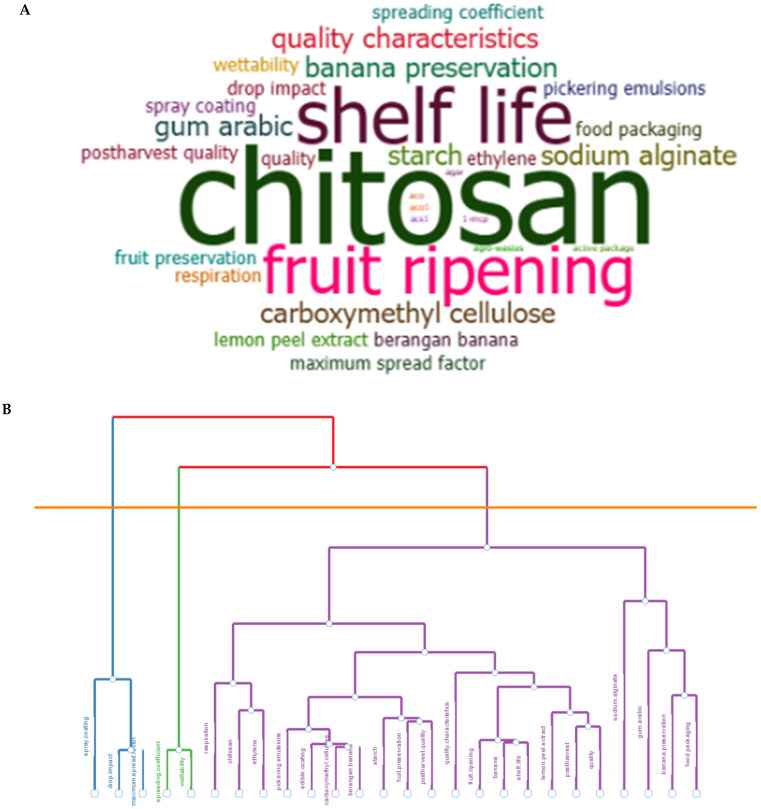
Overview of materials-based coatings used for banana fruit preservation (**A**) and schematic representation of the composition of the matrix of coated banana fruit (**B**).

**Figure 7 plants-14-00581-f007:**
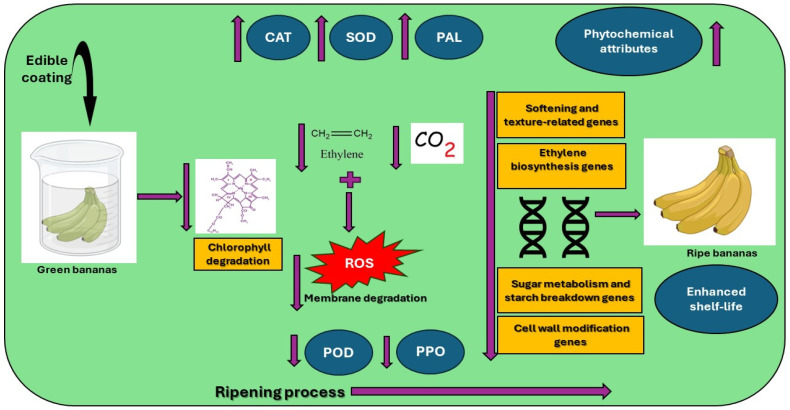
The mode of action of edible coatings on banana fruit during postharvest storage. This diagram shows coated fruit with suppressed (arrows down) chlorophyll degradation, ethylene and carbon dioxide production, oxidative stress, membrane degradation, peroxidase (POD), polyphenol oxidase (PPO), softening and texture-related genes, ethylene biosynthesis genes, sugar metabolism, and starch breakdown genes and cell wall modification genes, along with activated (arrows up) catalase (CAT), superoxide dismutase (SOD), phenylalanine ammonia-lyase (PAL), phytochemical attributes (phenolics, flavonoids, ascorbic acids, etc.), leading to delayed color change and enhanced shelf-life.

**Table 1 plants-14-00581-t001:** Most relevant journals with their respective articles, cite scores, and impact factors.

Journals	Articles	Cite Score	Impact Factor
*International Journal of Biological Macromolecules*	3	13.7	8.2
*Journal of Food Processing and Preservation*	3	2.2	2.6
*LWT*	3	11.8	6
*Postharvest Biology and Technology*	3	12	7
*International Food Research Journal*	2	12.5	8.1

**Table 2 plants-14-00581-t002:** Types of edible coatings materials with their respective occurrence in author’s keywords.

Classes	Materials	Number of Articles
**Polysaccharide-based coatings (71.8%)**	Agar–agar	1
	*Aloe vera* gel	4
	Carboxymethyl cellulose (CMC)	7
	Carrageenan	2
	Chitosan	19
	Guar gum	1
	Gum Arabic	6
	Hydroxypropyl methylcellulose	2
	Maltodextrin	1
	Methylcellulose	1
	Mucilage	1
	Pectin	2
	Sodium alginate	3
	Starch	5
	Tragacanth gum	1
**Protein-based coatings (10.3%)**	Fibroin	2
	Gelatin	5
	Whey protein	1
**Lipid-based coatings (1.2%)**	Beeswax	1
**Composite-based coatings (16.7%)**	Aloe vera–starch–gum Arabic	1
	Chitosan–gum Arabic	3
	Chitosan–pectin	1
	Chitosan–sodium alginate	1
	Chitosan–starch	1
	CMC–gelatin	1
	Guar gum–Aloe vera	1
	Guar gum–CMC–pectin	1
	Maltodextrin–gum Arabic–sodium alginate	1
	Starch–beeswax	1
	Starch–gelatin	1

**Table 3 plants-14-00581-t003:** Effect of edible coatings on physiological response and physicochemical and phytochemical attributes in postharvest banana fruit.

Edible Coatings	Coating Formulations	Storage Conditions	Key Findings	Reference
**Polysaccharide-based edible coatings on banana**				
Chitosan	1.25% CH and 1% + acetic acid	27 ± 0.7 °C and 82.57 ± 3.62% 85 ± 2% RH	No significant difference in weight loss and TSS. Delayed the changes in the pulp-to-peel ratio.	[52]]
	1.15% CH, 1.25% CH + CHNP	25 ± 1 °C	Delayed weight loss, suppressed TSS accumulation. Coated samples showed extended shelf-life.	[100]
	CH (1 and 1.5%), CaCl_2_ (1 and 1.5%), 1% CH + 100 ppm GBA, 1.5% CH + 100 gibberellic acid, 98% glycerol and jojoba wax	34 ± 1 °C and 70–75% RH	Delayed fruit weight reduction, sugar accumulation, and pigment degradation. Maintained ascorbic acid, TSS, and TA levels.	[53]
	1% acetic acid, CH (0.5 and 1%), TR (1.6 × 10^−5^; 1.6 × 10^−4^ and 1.6 × 10^−3^) and GB (10, 15 and 20 mM) and Tween-20 was added to all treatments	18 ± 2 °C and 60–70% RH	Did not significantly affect weight loss. Preserved peel color index, membrane stability index, firmness, and ascorbic acid content. Retained higher TA, TPC, and TFC. Improved FRSC (DPPH IC_50_).	[55]
	10 g/L CH, 20 μM EBR, and 10 g/L CH + 20 μM EBR + 1 mL/L tween 20 in all formulations.	23 ± 1 °C and 85–95% RH	Significantly delayed fruit ripening by delaying color change. Delayed weight loss and TSS accumulation. Retained firmness, TA, and membrane stability index. Retained higher radical scavenging capacity, TPC, and TFC.	[139]
	1% HMW-CH (540 kDa), 1% MMW-CH (265 kDa) and 1% LMW-CH (65 kDa) + 0.1% tween 80 in all formulations.	25 ± 2 °C	Delayed CO_2_ and C_2_H_4_ production. Preserved high firmness. Retained high antioxidant activity.	[54]
Chitosan	1% CH, CH-PBLE2%, CH-SOE2%, CH-SOE1%-PBLE1% and CS-SOE1.4%-PBLE-0.6%	20 °C and 64% RH	Reduced respiration rate, and weight loss. Maintained TSS and TA quality attributes.	[140]
	1% CH + 0.6% PEG, 1% CH + 0.6% PEG + 0.5% CIN, 1% CH + 0.6% PEG + 1% CIN, 1% CH + 0.6% PEG + 1.5% CIN, 1	25 °C and 75% RH	Delayed CO_2_ production peak, weight loss, and sugar content loss. Retained firmness, color, TSS, and TA attributes.	[141]
CMC	0.5%, 1% and 1.5% CMC + 0.25% Tween 20 and 1% glycerol in all formulation	20 ± 1 °C and 85 ± 2% RH	Delayed peel color changes. Suppressed C_2_H_4_ and CO_2_ production. Suppressed the TSS increase and exhibited higher TA. Maintained higher firmness, protopectin, chelate-soluble pectin, sodium carbonate soluble pectin, cellulose, and hemicellulose, while water-soluble pectin was lower.	[9]
CMC	1.5% CMC + 2% glycerol+ 2% Tween-80	25 °C and 85–90% RH	Effectively reduced weight loss. Delayed color changes by suppressing chlorophyll degradation and ripening. Retained firmness and membrane integrity. Preserved pectin and sugar content.	[2]
CMC	1.5% CMC + 2% tween 80, 0.5% CMC + 0.5% MRE, 1.5% CMC + 1.5% MRE and 2.5% CMC + 2.5% MRE	13 ± 2 °C and 40% RH	Delayed weight loss and color change, retained firmness. Decreased CO_2_ production. Induced sucrose and total chlorophyll.	[69]
Agar–agar	0.5, 1 and 2 g/L of agar–agar	25 ± 2 °C	Significantly decreased weight loss, and firmness loss and maintained TSS content. Showed significantly fewer cracks and smooth surfaces.	[142]
*Aloe vera*	50% AV, 50% AV + LPE 5%, 50% AV + LPE 10%, 50% AV + LPE 15% and 10 g/L glycerol was added in every formulation.	23 ± 1 °C and 78% RH	Reduced weight loss, TSS accumulation, and retained organic acids. Maintained the content of ascorbic acid.	[75]
Gum Arabic	GA (5 and 10%), SA (1 and 2 mM), and 10% GA + 1 mM SA. 0.5 mL/L was added in all formulations.	20 ± 2 °C and 60–70% RH	Preserved TSS, TA, firmness, peel color, pH, and membrane stability index. Retained higher TPC, TFC, and FRSC.	[143]
Poly (vinyl alcohol)	PP, PVA + 1% extract, PVA + 5% extract.	25 °C ± 1 °C	Delayed TA reduction and TSS increment. Slowed down chlorophyll degradation.	[131]
OFIM	1%, 2% and 3% OFIM and 0.1% glycerol	23 ± 2 °C and 85 ± 2% RH	Significantly reduced C_2_H_4_ and CO_2_ production. Delayed weight loss, color change, firmness loss, TSS increase, and TA reduction. Delayed cell wall and chlorophyll degradation, carotenoid accumulation.	[10]
Carrageenan	0.5%, 1% and 1.5% Car + 0.5% glycerol in all treatment	20 °C and 26 °C	Delayed color change. Delayed TSS increment. Reduced weight loss.	[61]
**Polysaccharide-based edible coatings**				
Starch	3% rice starch + 1.5% 1-car + 2% FAEs and 1% glycerol	20 ± 2 °C and 20 ppm ethylene	Reduced weight loss, firmness loss, C_2_H_4,_ and CO_2_ production. Delayed chlorophyll degradation and color change. Coated bananas were extended for four more days.	[27]
Konjac glucomannan	KGM, K-NPs-5, K-NPs-10, K-NPs-15, and K-NPs-20	25 °C and 50% RH	Reduced weight loss, and CO_2_ production. Retained antioxidants, and effectively prolonged shelf life.	[144]
Poly (viny alcohol)	PAM, PAM-0.2AgNPs, PAM-0.4AgNPs, and PAM-0.6AgNPs	25 °C	Reduced weight loss, respiration rate, acidity loss, pH, and TSS and maintained a good appearance. Slowed down softening and delayed ripening.	[103]
**Lipid-based edible coatings on banana**				
Beeswax	Coating 1: 25 g beeswax +140 mL coconut oil + 50 mL sunflower oil Coating 2: 25 g beeswax 200 mL coconut oil + 50 mL sunflower oil	NS	Successfully prevented deterioration of fruit. Preserved Vitamin C. Improved the fruit appearance.	[88]
**Protein-based edible coatings on banana**				
Soybean protein isolate	6% SPI, 6% SPI + 1 mg/mL CIN and 6% SPI + 1 mg/mL CIN + 1 mol/L ZnONP + 0.1% Tween-80 in all formulations	25 °C and 40% RH	Maintained TSS and TA attributes. Reduced weight loss and firmness loss.	[32]
Gelatin	40% gelatin and 40% gelatin + 5% PGE-400	25 °C	Preserved high TA and reduced TSS increment rate. Improved fruit color and weight loss.	[3]
**Composite-based edible coatings on banana**				
Chitosan, gallic acid and chitosan gallate	1% acetic acid, CH (0.15 and 1%), gallic acid (0.075 and 0.15%) and CH-G (25, 50 and 75 mL/L) + 0.5 mL/L tween 20.	20 ± 2 °C and 60–70% RH	Delayed TSS accumulation, TA decline, color changes, and weight loss. Did not affect ascorbic acid, flavonoid contents, and FRSC (DPPH IC50 values). Preserved total phenolic content.	[91]
Sodium alginate and whey protein isolate	SDA, S12W8, and PE	25 °C and 50% RH	Significantly reduced weight loss. Maintained high firmness and acidity.	[145]
Chitosan and Polyvinyl alcohol	CH + PVA, 10 mM CH + PVA + OA, 15 mM CH + PVA + OA and 20 mM CH + PVA + OA	24 ± 1 °C and 58 ± 2% RH	Reduced weight loss and peel browning index. Significantly improved peel color hue angle. Maintained total phenols.	[146]
Chitosan and gum Arabic	1.0% CH + 10.0% GA, and 1.0% CH + 10.0% GA with ZnO of 0.1%, 0.3%, 0.5%, and 1.0%	35 °C and 54% RH	Maintained pulp firmness, TA, and sugars. Reduced weight loss.	[147]
Aloe vera and chitosan	50% AV, 10% MO, 50% AV + 2% CH, and 10% MO + 2% CH. AA (4%) was added to AV only. Glycerol (2%) was added to all formulations	18 ± 1 °C and 35–45% RH	Significantly reduced firmness loss, weight loss, CO_2,_ and C_2_H_4_ evolution. Retained total phenolic content, TSS, and peel color.	[99]
Chitosan and gum Arabic	1% CH + 10% GA, 1% CH + 10% GA + 0.05% CE, and 1% CH + 10% GA + 0.15% CE, 1% CH + 10% GA + 0.25% CE and 1% CH + 10% GA + 0.5% CE	25 °C and 70% RH	Effectively delayed firmness loss, weight loss, acidity decline, and color. Significantly retained total polyphenol and flavonoid content.	[30]
Aloe vera, starch, and gum Arabic	7.30–19.36% AV, 0.64–13.69% starch, and 0.59–2.40% GA	24 ± 2 °C and 80 ± 2%	Delayed firmness loss, weight loss, and TSS synthesis. Suppressed chlorophyll degradation and carotenoid accumulation.	[29]

CMC: carboxymethyl cellulose, OFIM: *Opuntia ficus-indica* mucilage, PVA: poly(vinyl alcohol), Car: carrageenan, GA: gum Arabic, AV: *Aloe vera*, MO: *Moringa oleifera*, GBA: gibberellic acid, MRE: *Morus alba* root extract, CIN: cinnamaldehyde, SPI: soybean protein isolate, KGM: Konjac glucomannan, SA: salicylic acid, SDA: sodium alginate, TR: trans-resveratrol, GB: glycine betaine, OA: oxalic acid, CE: Cleistocalyx extracts, SOE: *Sonneratia ovata* Bock extract, PBLE: Piper-betle. L extract, PAM: poly (vinyl alcohol)/agar/maltodextrin, EBR: 24-Epibrassinolide, CH-G: chitosan gallate, LMW: low molecular weight, MMW: medium molecular weight, HMW: high molecular weight, LPE: lemon leaf extract, C_2_H_4_: ethylene, CO_2_: carbon dioxide, TA: titratable acidity, TSS: total soluble solids, TPC: total phenolic content, TFC: total flavonoid content, FRSC: free radical scavenging capacity, DPPH: 1,1-diphenyl-2-trinitro phenylhydrazine. NS: not specified; RH: relative humidity.

**Table 4 plants-14-00581-t004:** The summary of the effectiveness of various edible coatings in controlling microorganisms on bananas during postharvest storage.

Edible Coating	Microorganisms	Effectiveness	Reference
Agar–agar	*Colletotrichum musae* and *Fusarium moniliforme*	Significantly reduced disease incidence and severity	[139]
Gum Arabic and chitosan	*Colletotrichum musae*	Significantly reduced disease incidence	[148]
Aloe vera gel and lemon peel extract	*Colletotrichum musae*	Suppressed disease incidence	[74]
CMC	Unknown	Significantly reduced	[9]
OFIM	Unknown	Significantly reduced	[10]
Aloe vera and chitosan	Unknown	Reduced	[98]

## Data Availability

All data used have been included in the article.

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
