# Peer review of "Recent Advancements and Trends in Postharvest Application of Edible Coatings on Bananas: A Comprehensive Review"

_plants, 2025, doi:10.3390/plants14040581_

Round 1

Reviewer 1 Report

Comments and Suggestions for Authors

This review appears to be a well-structured and insightful contribution to the field of postharvest banana preservation. It effectively integrates bibliometric analysis with technical discussions on edible coatings, particularly focusing on polysaccharide-based coatings like chitosan. The inclusion of bibliometric analysis using Scopus and VOS-viewer adds value by highlighting research trends, author contributions, and geographical distribution, which provide a broader context to the topic.  It critically evaluates the current state of the field.

However, the review could benefit from a deeper exploration of the economic feasibility of large-scale application and potential regulatory hurdles. It would also be interesting to see a comparison between chitosan and other polysaccharides in terms of effectiveness, cost, and consumer acceptance.

Overall, this review seems to be a comprehensive and well-researched piece that synthesizes existing literature while providing insights into future advancements in edible coating technologies for bananas.

For that reason, I would accept this manuscript after minor revisions inclusion. 

Author Response

Authors’ responses: We appreciate your positive feedback on our manuscript and your meaningful suggestion about the economic feasibility, regulation, and scalability of edible coatings for bananas. We acknowledge these aspects will value our study; unfortunately, there is no available information on them in the literature, as we have already mentioned in section 9, “Limitations and future research” of our manuscript. However, based on your suggestion, we have refined the discussion by adding more clarifications.

Regarding the suggestion of integrating the comparison between chitosan and other materials (polysaccharides) in terms of effectiveness, cost, and consumer acceptance, we believe that, while important, it falls outside the primary focus of our current review. However, we recognize its significance and will consider addressing it in future work.

Reviewer 2 Report

Comments and Suggestions for Authors

Dear authors,

I would like to congratulate you on your choice of topic for this review, as I believe that it can be of important contribution in the field, at this point.

Minor revisions, in my opinion, are needed regarding the following:

1. Please reconsider the statement from the Abstract section:

"Despite their advantages as biodegradable, cost-effective alternatives to synthetic fungicides, the commercial application of edible coatings faces limitations, including scalability and practicality.

2. Please reconsider Key words order:

 bibliometric analysis, edible coatings, banana, active ingredient, ripening, mode of action

3. Section 4.1. L44-45: affordability of chitosan is very debatable

4. Section 4.1. L59-60: oil free starch? What did you mean? Other polysaccharides are not oil free?

5. Section 4.1. L62-66: Please separate it into a separate paragraph

6. Section 4.1. L75-76: please rewrite the sentence

7. Section 4.2. L113-115: The same as in polysaccharide. It is misleading to emphasize this only for protein-based coatings.

8. Section 4.2. L117-119: Please reconsider there are untrue and unclear statements

9. Section 4.2. L120-122: This sentence does not belong here, maybe at the beginning of section 4.2.

10. Section 4.4. L147: not "also known", but "or"

11. Section 5 L. 170: coating thickness

12. Section 5.5.1. L. 182: with lemon peel extract

13. Section 5.5.2. L. 190: anti-corrosion? Please explain

14. Section numbers are not good

15. Section 4.5.5. L. 278-280: Please separate it into a separate paragraph

16. Section 5. L. 284-300 Repeating, this was already discussed

Author Response

Reviewer 2: Dear authors,

I would like to congratulate you on your choice of topic for this review, as I believe that it can be of important contribution in the field, at this point.

Minor revisions, in my opinion, are needed regarding the following:

Comment 1. Please reconsider the statement from the Abstract section:

"Despite their advantages as biodegradable, cost-effective alternatives to synthetic fungicides, the commercial application of edible coatings faces limitations, including scalability and practicality.

Authors’ response: Thank you for your suggestion, we rephrased the sentence as suggested.

Comment 2. Please reconsider Key words order: bibliometric analysis, edible coatings, banana, active ingredient, ripening, mode of action

Authors’ response: Thank you for your suggestion, we re-arranged the keywords as suggested.

Comment 3. Section 4.1. L44-45: affordability of chitosan is very debatable

 Authors’ response: Thank you for your comment, we have removed the word. L330

Comment 4. Section 4.1. L59-60: oil free starch? What did you mean? Other polysaccharides are not oil free?

  Authors’ response: Thank you for your comment. We have removed the term “oil free” and added more information starch-based coating to enhance the manuscript. L345-346

Comment 5. Section 4.1. L62-66: Please separate it into a separate paragraph

 Authors’ response: Thank you for your comment, we have separated the paragraphs as suggested. L349-350

Comment 6. Section 4.1. L75-76: please rewrite the sentence

 Authors’ response: Thank you for your comment, we have rephrased the suggested sentence to enhance clarity. L362-363

Comment 7. Section 4.2. L113-115: The same as in polysaccharide. It is misleading to emphasize this only for protein-based coatings.

 Authors’ response: Thank you for your comment, we have removed the statement as suggested.

Comment 8. Section 4.2. L117-119: Please reconsider there are untrue and unclear statements

 Authors’ response: Thank you for your comment, regarding the ‘unclear” statement, we have rephrased the sentence to improve clarity (L404-406). Regarding the ‘untrue’ statement, we want to clarify that the statement was adopted from the published article, which we have appropriately cited. Additionally, we have attached the article doi “https://doi.org/10.1016/j.tifs.2018.03.003’ which stated that “Proteins-based coatings usually display high gas permeability, good mechanical properties, and low moisture barriers “

Comment 9. Section 4.2. L120-122: This sentence does not belong here, maybe at the beginning of section 4.2.

 Authors’ response: Thank you for your suggestion, we have re-positioned the sentence as suggested (398-400)

Comment 10. Section 4.4. L147: not "also known", but "or"

  Authors’ response: Thank you for your correction, we have fixed the matter. L431

Comment 11. Section 5 L. 170: coating thickness

Authors’ response: Thank you for your correction, we have fixed the wording. L454

Comment 12. Section 5.5.1. L. 182: with lemon peel extract

Authors’ response: Thank you for your correction, we have inserted the word as suggested (L466)

Comment 13. Section 5.5.2. L. 190: anti-corrosion? Please explain

 Authors’ response: Thank you for your comment, we acknowledge the error and appreciate the reviewer’s attention to detail. The term “anti-corrosion” was mistakenly included, and we have now corrected it by replacing it with “anti-inflammatory properties, which accurately reflects the intended context (L474).

Comment 14. Section numbers are not good

 Authors’ response: Thank you for your observation, we have now corrected the section numbers accordingly.

Comment 15. Section 4.5.5. L. 278-280: Please separate it into a separate paragraph

 Authors’ response: Thank you for your suggestion. The paragraphs were separated as suggested. L563-565

Comment 16. Section 5. L. 284-300 Repeating, this was already discussed

Authors’ response: Thank you for your comment, we have rephrased the paragraph intending to introduce Table 3. L569-576

Reviewer 3 Report

Comments and Suggestions for Authors

This paper (review) studies edible coating technologies for banana preservation. This paper is characterized by a very deep and up-to-date statistical study of the bibliography on the subject. It also covers pros and cons of edible coating technologies for banana preservation, as well as coating production technology. One aspect to highlight is the applied didactics with very concrete explanatory figures and tables that will make the reader very attracted to this work. This is a good review to be published in "Plants".

Author Response

Reviewer 3: This paper (review) studies edible coating technologies for banana preservation. This paper is characterized by a very deep and up-to-date statistical study of the bibliography on the subject. It also covers pros and cons of edible coating technologies for banana preservation, as well as coating production technology. One aspect to highlight is the applied didactics with very concrete explanatory figures and tables that will make the reader very attracted to this work. This is a good review to be published in "Plants".

 Authors’ response: We appreciate the reviewer’s positive feedback on our work. We are pleased that the statistical analysis, discussion of edible coating technologies, and the inclusion of explanatory figures and tables were well received. Our goal was to provide a comprehensive and accessible review, and we are grateful for the reviewer’s recognition of its contribution to the field.